# Gut Microbiota and Non-Alcoholic Fatty Liver Disease Severity in Type 2 Diabetes Patients

**DOI:** 10.3390/jpm11030238

**Published:** 2021-03-23

**Authors:** Hui-Ju Tsai, Yi-Chun Tsai, Wei-Wen Hung, Wei-Chun Hung, Chen-Chia Chang, Chia-Yen Dai

**Affiliations:** 1Department of Family Medicine, Kaohsiung Municipal Ta-Tung Hospital, Kaohsiung Medical University Hospital, Kaohsiung Medical University, Kaohsiung 801, Taiwan; bankin_0920@yahoo.com.tw; 2Department of Family Medicine, School of Medicine, College of Medicine, Kaohsiung Medical University, Kaohsiung 807, Taiwan; 3Division of General Medicine, Kaohsiung Medical University Hospital, Kaohsiung Medical University, Kaohsiung 807, Taiwan; lidam65@yahoo.com.tw; 4Division of Nephrology, Kaohsiung Medical University Hospital, Kaohsiung Medical University, Kaohsiung 807, Taiwan; 5School of Medicine, College of Medicine, Kaohsiung Medical University, Kaohsiung 807, Taiwan; 6Cohort Research Center, Kaohsiung Medical University, Kaohsiung 807, Taiwan; 7Division of Endocrinology and Metabolism, Kaohsiung Medical University Hospital, Kaohsiung Medical University, Kaohsiung 807, Taiwan; hung4488@ms57.hinet.net; 8Department of Microbiology and Immunology, College of Medicine, Kaohsiung Medical University, Kaohsiung 807, Taiwan; wchung@kmu.edu.tw (W.-C.H.); wacsun@gmail.com (C.-C.C.); 9Department of Internal Medicine, Division of Hepatobiliary, Kaohsiung Medical University Hospital, Kaohsiung Medical University, Kaohsiung 807, Taiwan

**Keywords:** gut microbiota, type 2 diabetes, non-alcoholic fatty liver disease, fibroscan

## Abstract

Introduction: Non-alcoholic fatty liver disease (NAFLD) remains an important health issue worldwide. The increasing prevalence of NAFLD is linked to type 2 diabetes (T2D). The gut microbiota is associated with the development of NAFLD and T2D. However, the relationship between gut microbiota and NAFLD severity has remained unclear in T2D patients. The aim of this study was to evaluate the relationship of gut microbiota with the severity of NAFLD in T2D patients. Methods: This cross-sectional study used transient elastography (FibroScan) to evaluate the severity of hepatic steatosis. We utilized qPCR to measure the abundance of *Bacteroidetes*, *Firmicutes*, *Faecalibacterium prausnitzii*, *Clostridium leptum* group, *Bacteroides*, *Bifidobacterium*, *Akkermansia muciniphila*, and *Escherichia coli*. Results: Of 163 T2D patients, 83 with moderate to severe NAFLD had higher abundance of bacteria of the phylum *Firmicutes* with respect to 80 patients without NAFLD or with mild NAFLD. High abundance of the phylum *Firmicutes* increased the severity of NAFLD in T2D patients. A positive correlation between NAFLD severity and the phylum *Firmicutes* was found in T2D male patients with body mass index ≥24 kg/m^2^ and glycated hemoglobin <7.5%. Conclusion: Enrichment of the fecal microbiota with the phylum *Firmicutes* is significantly and positively associated with NAFLD severity in T2D patients. The gut microbiota is a potential predictor of NAFLD severity in T2D patients.

## 1. Introduction

Non-alcoholic fatty liver disease (NAFLD) remains an important health issue worldwide [1]. In Taiwan, where chronic hepatitis B and hepatitis C are prevalent, NAFLD is one of the most common liver diseases after advances in antiviral agents for viral hepatitis have been made [2]. NAFLD is defined by the pathologic accumulation of fat in the liver, is the most common cause of chronic liver diseases [3], and further progresses to liver cirrhosis and hepatocellular carcinoma [4]. NAFLD is a complex disease resulting from the interactions among polygenic and environmental factors. The increasing prevalence of NAFLD is related to metabolic diseases, such as type 2 diabetes (T2D), obesity, hypertension, renal dysfunction, and dyslipidemia [5]. Approximately 40–50% of T2D patients have NAFLD [6]. NAFLD and T2D share some common pathophysiologic mechanisms, including insulin resistance, genetic predisposition, and influence of environmental factors [7,8]. 

Recently, increasing evidence has indicated that the gut microbiota is involved in the development of NAFLD and T2D [9,10,11,12]. The gut microbiota establishes a complex network regulating the intestinal barrier with the immunology and enteroendocrine systems [9,13]. The gut microbiota is considered a microbial organ with specific physical functions [14]. Dysbiosis leads to the alteration of the intestinal barrier permeability and increased secretion of endotoxins, impairing insulin resistance and inducing inflammation [15,16,17]. More than 90% prevalent bacterial species can be grouped into two bacterial phyla, i.e., *Bacteriodetes* (Gram-negative) and *Firmicutes* (Gram-positive) [18], which play a principal role in the pathophysiology of metabolic disorders [1]. Patients with diabetes usually have higher abundance of the phylum *Bacteriodetes* and lower abundance of the phylum *Firmicutes* than do patients without diabetes [13]. *Faecalibacterium prausnitzii*, belonging to the *Clostridium leptum* group, one genus of the phylum *Firmicutes*, is the most abundant bacterium in the intestinal microbiota of healthy adults, accounting for more than 5% of the total bacterial population [13], but its abundance is decreased in T2D patients. Lower concentrations of bacteria of the genus *Bacteroides*, belonging to the phylum *Bacteriodetes*, and higher concentrations of bacteria of the of genus *Bifidobacterium*, belonging to the phylum *Actinobacteria*, have been found in T2D patients [19,20]. *Akkermansia muciniphila* has been reported to improve glucose tolerance and adipose tissue inflammation [21,22]. A large study using metagenomic sequencing in Chinese individuals reported increased abundance of *Escherichia coli*, belonging to the phylum *Proteobacteria*, in T2D patients [23]. Previous animal reports have indicated the role of gut microbiota in the pathogenesis in NAFLD [24,25]. However, the abundance of the gut microbiota in NAFLD is still discordant and poorly investigated in human studies [26,27,28]. Besides, the impact of the gut microbiota on the severity of NAFLD remains unclear, especially in patients with T2D. Therefore, this study was aimed to evaluate the relationship between the severity of NAFLD and the abundance of targeted fecal bacterial species, including the phyla *Bacteriodetes* and *Firmicutes*, the *C. leptum* group, the genera *Bacteroides* and *Bifidobacterium, F*. *prausnitzii, A. muciniphila*, and *E. coli* in T2D patients.

## 2. Materials and Methods

### 2.1. Study Participants

This observational study was conducted at a tertiary hospital in Southern Taiwan. We screened 203 patients with T2D from October 2016 to December 2017. We excluded patients with T2D having hepatitis B, hepatitis C, autoimmune hepatitis, alcohol consumption >30 g per day on average, and using antibiotics and probiotic or prebiotic products before enrollment in this study, and finally, 180 patients entered this study (Figure 1). This research was approved by the Institutional Review Board of Kaohsiung Medical University Hospital (KMUHIRB-G(II)-20160021). All patients signed an informed consent, and all clinical investigations were conducted on the basis of the principles expressed in the Declaration of Helsinki.

### 2.2. Clinical Data and Sample Collection

T2D was defined as a history of diabetes or the use of antihyperglycemic therapy, and according to serum sugar levels in relation to the American Diabetes Association criteria. We collected demographic characteristics from patients’ interview and medical records. Detailed information on the use of medications including antihyperglycemic and antidyslipidemic therapy was obtained from medical records. All study subjects were recruited during a diabetes education program and followed the principles of dietary treatment to T2D individually. We collected usual diet habits in these patients using a simple questionnaire. Body mass index (BMI) was calculated as body weight divided by body height squared. The patients were asked to fast for at least 12 h before blood and urine specimen collection for biochemistry studies and measurement of albuminuria respectively.

### 2.3. Transient Elastography

Transient elastography (TE) is an ultrasound-based method of elastography, which can evaluate hepatic steatosis by measuring the controlled attenuation parameter (CAP), with strong correlation to the stage of hepatic steatosis examined by simultaneous liver biopsy [29,30,31]. CAP was assessed simultaneously using a FibroScan (Echosens) with the standard 3.5 MHz M probe in the same cylinder of liver parenchyma (1 × 4 cm) at enrollment. CAP was a measure of ultrasonic attenuation at 3.5 MHz on the FibroScan signal and was expressed as dB/m [32]. We defined no NAFLD as CAP ≤ 238 dB/m, mild NAFLD as CAP within 239~258 dB/m, moderate NAFLD for CAP within 259–289 dB/m, and severe NAFLD for CAP ≥ 290 dB/m (https://www.mskcc.org/cancer-care/patient-education/understanding-your-fibroscan-results#section-2 accessed on 20 March 2020).

### 2.4. Stool Sample Collection and Microbial DNA Extraction for Real-Time Quantitative Polymerase Chain Reaction (qPCR)

The detailed methods were described previously [33]. In brief, fecal samples were collected at the same day of blood and urine collection. The excrement samples were collected by participants at home and were instantly frozen at −4 °C before bringing the samples to the hospital. Then, the excrement specimens were deposited at −80 °C before measurement. We extracted bacterial DNA using the Stool DNA Extraction kit (Topgen Biotechnology Co., Ltd., Taiwan). The weight of stool specimens was 50 to 100 mg, determined after bead beating (45 s; 3450 oscillations/min). After examining the quality and concentration of DNA, the extracted DNA specimens were placed at −20 °C before processing.

We used real-time qPCR in a StepOnePlus Real-Time PCR system (Thermo Fisher Scientific, Waltham, MA, USA) to detect bacterial 16S rRNA gene copies in stool according to a previous study [26,33]. The 8 pairs of 16S rRNA gene primers specific to *Firmicutes*, *Bacteroides*, the *C. leptum group*, *F. prausnitzii*, *Bifidobacterium*, *E. coli*, and *A. muciniphila* are listed in Appendix A. All qPCR tests were carried out twice, and the present data are the mean levels of repeated qPCR examination.

### 2.5. Statistical Analysis

Patients were stratified according to the severity of NAFLD (no–mild vs. moderate–severe). Continuous variables were expressed as mean ± SD or median (25th, 75th percentile), as appropriate, and categorical variables were expressed as percentages. Continuous variables with skewed distribution were log-transformed to approximate normal distribution. The significance of differences in continuous variables between groups was tested using independent t-test or Mann–Whitney U analysis, as appropriate. Differences in the distribution of the categorical variables were tested using the chi-square test. Multivariate logistic regression models were used to evaluate the association between the microbiota and the severity of NAFLD. All the variables in Table 1 examined by univariate analysis and those variables with *p*-value < 0.05, age, and gender were selected in multivariate logistic regression models. Statistical analyses were carried out by SPSS version 18.0 for Windows, and the graphs were made using GraphPad Prism 5.0 (GraphPad Software Inc., San Diego, CA, USA). Statistical significance was set at a two-sided *p*-value < 0.05.

## 3. Results

### 3.1. Characteristics of the Entire Cohort

Table 1 shows the comparison of the clinical features of groups based on the severity of NAFLD. After excluding 17 patients with diabetes presenting a *Firmicutes*/*Bacterodietes* ratio at the extremities of the normal range (<0.1 or >1.0), 163 patients were included in the final analysis (Figure 1). Eighty subjects with T2D had mild NAFLD or no NAFLD, and 83 had moderate to severe NAFLD. The study subjects had a mean age of 63.6 ± 9.8 years and had suffered from diabetes for 10.3 ± 7.2 years. Of 163 subjects, 22.1% had an alcohol habit, and 28.2% had a smoking habit. The prevalence of hypertension, gout, and hyperlipidemia was 68.1%, 8.6%, and 85.9%, respectively. The mean CAP was 265.2 ± 46.7 dB/m. T2D patients with moderate to severe NAFLD had higher BMI than those without NAFLD or with mild NAFLD. There was no difference in smoking, alcohol use, diet habits, anti-diabetic agent usage, or statin usage between the two groups. Higher glutamic pyruvic transaminase (GPT) and triglyceride levels and a lower high-density lipoprotein (HDL) level were found in T2D patients with moderate to severe NAFLD compared to patients without NAFLD or with mild NAFLD.

### 3.2. The Distribution of the Gut Microbiota in T2D Patients with Different Levels of Severity of NAFLD

We examined the phyla *Firmicutes* and *Bacterodietes*, the *C. leptum group*, the *genera Bacteroides* and *Bifidobacterium*, *A. muciniphila*, *F. prausnitzii*, and *E. coli* using qPCR in the study subjects. T2D patients with moderate to severe NAFLD had higher abundance of the phylum *Firmicutes* than those without NAFLD or with mild NAFLD. There was no difference in the abundance of the phylum *Bacterodietes*, the *C. leptum group*, the genera *Bacteroides* and *Bifidobacterium*, *A. muciniphila*, *F. prausnitzii*, and *E. coli* between the two groups (Table 2).

### 3.3. Gut Microbiota and Severity of NAFLD

To examine the correlation between gut microbiota and severity of NAFLD, we used logistic regression analysis. In univariate analysis, BMI, serum GPT, triglyceride levels, and the phylum *Firmicutes* were significantly and positively associated with increased risk of moderate to severe NAFLD in T2D patients. Serum HDL was negatively correlated with increased risk of moderate to severe NAFLD (Table 3). We further performed multivariate analysis adjusting for age, sex, BMI, serum GPT, triglyceride and HDL levels, and the results showed that T2D subjects with high abundance of the phylum *Firmicutes* (odds ratio (OR): 3.68, 95% confidence index (CI): 1.30–10.46) had increased risk of moderate to severe NAFLD.

To explore the effects of gender, obesity, and glycemic control on the correlation between the abundance of the phylum *Firmicutes* and the severity of NAFLD, we stratified the study subjects by sex, BMI of 24 kg/m^2^, and level of glycated hemoglobin (HbA1c) of 7.5% (Figure 2), and the results revealed a positive correlation between the phylum *Firmicutes* and the severity of NAFLD in T2D male patients with BMI ≥ 24 kg/m^2^ and HbA1c < 7.5%.

## 4. Discussion

This study evaluated the association of the gut microbiota with the severity of NAFLD in T2D patients. We found higher abundance of the phylum *Firmicutes* in T2D patients with moderate to severe NAFLD than in those without NAFLD or with mild NAFLD. Furthermore, after adjusting for metabolic factors, including age, sex, BMI, liver function, triglyceride and HDL–cholesterol levels, a high abundance of the phylum *Firmicutes* significantly increased the risk of greater severity of NAFLD in T2D patients. To our best knowledge, this is the first study to evaluate the gut microbiota in T2D patients with NAFLD. The gut microbiota might not only play a role in the pathophysiology of NAFLD, but also have an independent predictive value for the severity of NAFLD in T2D patients.

Previous studies have revealed that the gut microbiota is linked to the development of NAFLD in mice [24,25]. However, information concerning the relationship between the gut microbiota and the severity of NAFLD is limited. It is well known that obese individuals have increased levels of bacteria of the phylum Firmicutes and decreased levels of bacteria of the phylum *Bacterodietes*, and T2D patients have reduced phylum *Firmicutes* and elevated phylum *Bacterodietes* [15]. Larsen et al. also found that T2D patients had a significantly lower abundance of *Firmicutes* than healthy individuals [34]. A recent report conducted by Boursier et al. found an association of the genus *Bacteroides* with the severity of NAFLD in the European continent [26]. Conversely, no correlation between the genus *Bacteroides* and NAFLD severity was seen in a Thai population [27]. Different from these previous studies, we focused on T2D patients and enrolled more patients to evaluate the correlation between the gut microbiota and the severity of NAFLD. Interestingly, we found enrichment of the fecal phylum *Firmicutes* in T2D patients with moderate–severe NAFLD, which correlated with the severity of NAFLD, especially in obese T2D patients, differently from the results of Roman et al. and Wong et al. [35,36]. Our findings suggest that NAFLD may influence the composition of the gut microbiota in T2D patients, and the underlying mechanisms need further study to be identified. In addition to variations in age, sex distribution, race, dietary habit, and sample size, the complex interaction between obesity and diabetes might explain these inconsistencies. On the other hand, we used qPCR to measure the absolute value of microbial components instead of Next-Generation Sequencing (NGS), which provides the relative value of microbial components. This partially explains the different results between our study and other works.

In further subgroup analysis, the significant correlation between the fecal phylum *Firmicutes* and the severity of NAFLD was shown in male subjects with diabetes, not in female patients. On the other hand, female patients had a high risk of severe NAFLD compared to male patients. In the female group, there might be other causes leading to severe NAFLD instead of the gut microbiota. Besides, our results revealed that the fecal phylum *Firmicutes* was associated with the severity of NAFLD in T2D patients with HbA1c < 7.5% (good glycemic control), not in those with HbA1c ≥ 7.5% (poor glycemic control). The lower abundance of the phylum *Firmicutes* has been traditionally found in T2D patients compared to normal individuals [13]. However, the abundance of the phylum *Firmicutes* was not different between T2D patients with HbA1c < 7.5% and those with HbA1c ≥ 7.5%, and there was no association of the abundance of the phylum *Firmicutes* with the HbA1c level. Further study is necessary to evaluate the effect of the phylum *Firmicutes* in enhancing the severity of NAFLD in T2D patients with different HbA1c levels.

The pathophysiological mechanisms through which the gut microbiota may mediate NAFLD severity are not well explored. The colonic microbiota induces the fermentation of nondigestible carbohydrates leading to the formation of short-chain fatty acids (SCFAs) [37]. The net effect of SCFAs on the pathophysiology of NAFLD is intricate and remains unclear [38,39,40]. SCFAs include three major components, i.e., acetate, propionate, and butyrate. Butyrate, known to increase insulin sensitivity in mice and having an anti-inflammatory effect [41], is the predominant product of the phylum *Firmicutes* through carbohydrate fermentation. Acetate and propionate are mainly produced by the phylum *Bacteroidetes* [13,19]. Acetate acts as the substrate for cholesterol synthesis, and propionate participates in gluconeogenesis through the portal circulation.

Although liver biopsy is the gold standard for the diagnosis of NAFLD, we used Fibroscan to identify the severity of NAFLD in the study. Fibroscan is a non-invasive device based on the technique of TE to evaluate the stiffness of liver. Previous studies suggested that the reliability of Fibroscan was better than that of other non-invasive tools, and Fibroscan was validated with respect to liver biopsy [12,42]. Eddowes et al. used Fibroscan to assess the 450 patients with the complete spectrum of NAFLD fibrosis stages and found the good area under the receiver operating characteristic values ranged from 0.70 to 0.89 for both fibrosis and steatosis [42]. Therefore, Fibroscan is a convenient and reliable tool to diagnose NAFLD for clinical physicians.

This study has several limitations. An alteration of the microbial composition causes imbalances in butyrate, acetate, and propionate proportions, leading to abnormal metabolism. However, we did not measure SCFAs in the feces, and this is one of the limitations in the current study. An increase in the phylum Firmicutes is correlated with the risk of severe NAFLD, thereby we hypothesize that an imbalance in the abundance of the phylum *Firmicutes* might disturb SCFA production, affecting SCFA types and proportions and resulting in NAFLD onset and progression. Further study is necessary to examine the relationship between SCFA and the gut microbiota in T2D patients with NAFLD. Another limitation is the relatively small number of study subjects enrolled in this study. However, we still found a strong relationship between the gut microbiota and NAFLD severity in T2D patients. This study reveals the potential impact of the gut microbiota on NAFLD severity, consequently providing more precise clues about why T2D patients are at high risk for severe NAFLD.

## 5. Conclusions

In conclusion, enrichment of the fecal microbiota with the phylum *Firmicutes* is significantly and positively associated with NAFLD severity in T2D patients. In addition to traditional metabolic factors, the gut microbiota has the potential to predict NAFLD severity.

## Figures and Tables

**Figure 1 jpm-11-00238-f001:**
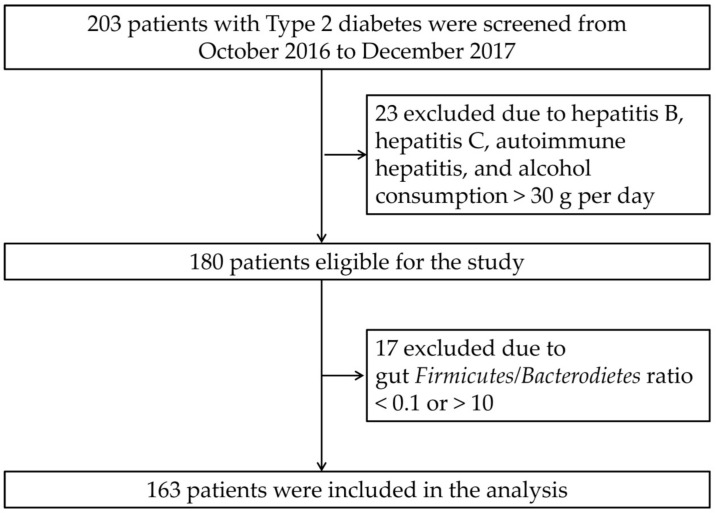
Flowchart.

**Figure 2 jpm-11-00238-f002:**
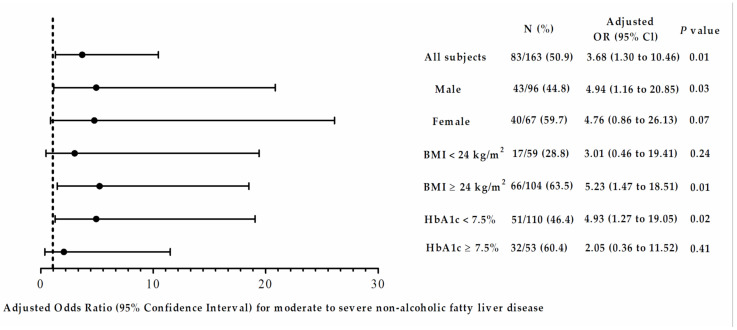
Adjusted odds ratios (ORs) of the severity of non-alcoholic fatty liver disease in patients with T2D stratified by sex, body mass index cut at 24 kg/m^2^, and glycated hemoglobin (HbA1c) cut at 7.5%. Ratios were adjusted for age, sex, body mass index, glutamate pyruvate transaminase (GPT), log-formed triglyceride, and high-density lipoprotein.

**Table 1 jpm-11-00238-t001:** The characteristics of the study participants.

	Entire Cohort (*n* = 163)	No or Mild NAFLD (*n* = 80)	Moderate or Severe NAFLD (*n* = 83)	*p*-Value
Age, year	63.6 ± 9.8	65.3 ± 9.5	62.0 ± 9.9	0.03
Sex (male), %	58.9	66.3	51.8	0.06
Smoke, %	28.2	25.0	31.3	0.37
Alcohol, %	22.1	20.0	24.1	0.53
Hypertension, %	68.1	63.8	72.3	0.24
Gout, %	8.6	8.8	8.4	0.94
Dyslipidemia, %	85.9	81.3	90.4	0.09
Diabetes duration, year	10.3 ± 7.2	10.8 ± 7.5	9.9 ± 6.9	0.44
Body Mass Index, kg/m^2^	25.6 ± 3.5	24.1 ± 2.7	26.9 ± 3.5	<0.001
CAP, dB/m	265.2 ± 46.7	226.4 ± 20.6	302.5 ± 32.3	<0.001
Diet habit, %				0.15
Protein more than fiber	13.6	10.8	16.3	
Fiber more than protein	30.5	37.8	23.8	
Fiber equal to protein	55.8	51.4	60.0	
Medication				
Sulfourea (yes vs. no)	51.5	51.3	51.8	0.94
DPP4 inhibitor (yes vs. no)	68.7	70.0	67.5	0.72
Metformin (yes vs. no)	84.0	86.3	81.9	0.45
Pioglitazone (yes vs. no)	3.7	5.0	2.4	0.38
Insulin (yes vs. no)	19.6	13.8	25.3	0.06
Statin (yes vs. no)	64.4	57.5	71.1	0.07
Laboratory parameters				
Cr, mg/dL	1.0 ± 0.5	1.0 ± 0.4	1.0 ± 0.5	0.59
Hemoglobin, g/dL	13.5 ± 1.6	13.3 ± 1.7	13.7 ± 1.4	0.28
Albumin, g/dL	4.5± 0.4	4.5 ± 0.4	4.6 ± 0.3	0.12
Uric acid, mg/dL	5.8 ± 1.5	5.8 ± 1.4	5.9 ± 1.6	0.59
GOT	28.1 ± 11.2	26.8 ± 10.9	29.3 ± 11.5	0.17
GPT	28.6 ± 16.3	24.6 ± 13.8	32.3 ± 17.6	0.002
Cholesterol, mg/dL	163.1 ± 35.8	161.9 ± 37.9	164.2 ± 33.8	0.69
Triglyceride, mg/dL	113 (77, 163)	93 (65, 140)	133 (100, 186)	<0.001
HDL, mg/dL	44.9 ± 11.9	47.2 ± 13.3	42.6 ± 10.2	0.02
LDL, mg/dL	88.1 ± 27.1	86.4 ± 28.2	89.7 ± 26.1	0.43
HbA1c, %	6.9 (6.5,7.8)	6.8 (6.4, 7.5)	7.0 (6.5, 8.0)	0.04
Urine ACR	16.4 (6.7, 60.3)	17.8 (7.0, 49.1)	16.3 (6.4, 69.3)	0.98
C-reactive protein	1.0 (0.5, 2.0)	1.1 (0.6, 2.1)	0.9 (0.5, 2.0)	0.36

Abbreviations: NAFLD, non-alcoholic fatty liver disease; CAP, controlled attenuation parameter, LS, liver stiffness, DPP4, dipeptidyl peptidase 4; ACEI, angiotensin-converting enzyme inhibitors; ARB, angiotensin II receptor blockers; Cr, creatinine; GOT, glutamate oxaloacetate transaminase; GPT, glutamate pyruvate transaminase; HDL, high-density lipoprotein; LDL, low-density lipoprotein; ACR, albumin/creatinine ratio.

**Table 2 jpm-11-00238-t002:** Microbiota distribution in the study participants.

Microbiome	Entire Cohort (*n* = 163)	No or Mild NAFLD (*n* = 80)	Moderate or Severe NAFLD (*n* = 83)	*p*-Value
*Firmicutes*, copies *10^9^/g	5.4 (2.7, 8.8)	4.5 (2.3, 7.0)	6.3 (3.2, 11.1)	0.01
*Bacteroidetes*, copies *10^9^/g	7.9 (3.7, 17.2)	7.7 (2.5, 1.8)	8.4 (4.2, 16.3)	0.37
*Firmicutes/Bacteroidetes*	0.7 (0.4, 1.3)	0.6 (0.3, 1.7)	0.7 (0.4, 1.2)	0.54
*Clostridium leptum group*, copies *10^8^/g	7.1 (2.9, 15.2)	5.5 (2.5, 11.9)	7.9 (3.7, 17.7)	0.10
*Faecalibacterium prausnitzii*, copies *10^7^/g	10.4 (2.4, 36.4)	7.8 (2.7, 28.5)	14.6 (2.3, 49.9)	0.40
*Bacteroides*, copies *10^9^/g	1.8 (0.9, 3.9)	1.5 (0.8, 3.9)	2.0 (1.0, 3.9)	0.29
*Bifidobacterium*, copies *10^6^/g	3.5 (0.4, 18.4)	3.4 (0.4, 10.2)	3.8 (0.4, 26.7)	0.24
*Akkermansia muciniphila*, copies *10^4^/g	1.4 (0.3, 999.1)	3.5 (0.4, 852.2)	0.9 (0.3, 1812.9)	0.31
*Escherichia coli*, copies *10^8^/g	1.0 (0.2, 5.9)	1.1 (0.3, 5.4)	0.7 (0.2, 6.3)	0.57

**Table 3 jpm-11-00238-t003:** Logistic regression of determinants of the severity of NAFLD.

Moderate to Severe NAFLD	Crude OR (95% Cl)	*p*-Value	Adjusted OR (95% Cl)	*p*-Value
Clinical data				
Age, year	0.97 (0.93–0.99)	0.03	0.99 (0.95–1.03)	0.73
Sex (female vs. male)	1.68 (0.89–3.20)	0.11	2.57 (1.17–5.67)	0.02
Body mass index, kg/m^2^	1.34 (1.19–1.50)	<0.001	1.28 (1.12–1.46)	<0.001
Smoke (yes vs. no)	1.37 (0.69–2.72)	0.37	-	-
Alcohol (yes v.s. no)	1.27 (0.60–2.67)	0.53	-	-
Diet habit, %				
Fiber more than protein	0.42 (0.15–1.20)	0.11		
Laboratory data				
Creatinine, mg/dL	0.84 (0.45–1.56)	0.59	-	-
Hemoglobin, g/dL	1.13 (0.93–1.36)	0.22	-	-
Albumin, g/dL	2.05 (0.82–5.11)	0.13	-	-
Uric acid, mg/dL	1.06 (0.86–1.29)	0.59	-	-
GOT	1.02 (0.99–1.05)	0.17	-	-
GPT	1.04 (1.01–1.06)	0.004	1.01 (0.98–1.03)	0.49
Cholesterol, mg/dL	1.00 (0.99–1.01)	0.68	−	−
Log (Triglyceride)	12.12 (3.14–46.82)	<0.001	5.18 (0.91–29.65)	0.07
HDL, mg/dL	0.97 (0.94–0.99)	0.02	0.99 (0.95–1.02)	0.53
LDL, mg/dL	1.01 (0.99–1.01)	0.44	-	-
HbA1c, %	1.26 (0.98–1.60)	0.07	-	-
Log (urine ACR)	0.97 (0.66–1.41)	0.86	-	-
Log (C-reactive protein)	0.68 (0.31–1.47)	0.32	-	−
Microbiome				
Log(*Firmicutes*/g)	2.71 (1.13–6.48)	0.02	3.68 (1.30–10.46)	0.01
Log (*Bacteroidetes/g*)	1.52 (0.86–2.67)	0.14	-	-
Log (*Firmicutes/Bacteriodetes*)	1.02 (0.51–2.04)	0.95	-	-
Log (*C. leptum group*/g)	1.56 (0.86–2.85)	0.14	-	-
Log (*Bacteroides*/g)	1.39 (0.80–2.42)	0.24	-	-
Log (*Bifidobacterium*/g)	1.14 (0.92–1.41)	0.23	-	-
Log (*F. prausnitzii*/g)	0.97 (0.72–1.31)	0.84	-	-
Log (*A. muciniphila*/g)	0.94 (0.80–1.08)	0.37	-	-
Log (*E.coli*/g)	0.93 (0.69–1.24)	0.60	-	-

Abbreviations: OR, odds ratio; DPP4, dipeptidyl peptidase 4; GOT, Glutamate oxaloacetate transaminase; GPT, Glutamate pyruvate transaminase.

## Data Availability

The data presented in this study are available on request from the corresponding author. The data are not publicly available due to privacy.

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
