# Peer review of "Gut Microbiota and Non-Alcoholic Fatty Liver Disease Severity in Type 2 Diabetes Patients"

_jpm, 2021, doi:10.3390/jpm11030238_

Round 1

Reviewer 1 Report

Authors performed a cross-sectional study on 163 patients investigating the association between gut microbiota and NAFLD severity in patients with diabetes mellitus type 2.

The methods are well described and the results clear, I have no requests to make to the Authors

Author Response

Thanks for your comments

Reviewer 2 Report

Dear Authors,

I have read with great interest your article, and in order to improve its quality here are my suggestions:

  1. Study limitations are listed clearly in the manuscript. However, I would suggest specifying study limitations in a separate paragraph.
  2. English language and style are fine, however minor spell check is required. In the manuscript, I would suggest to change following terms:
  • instead using - Type 2 diabetes mellitus (T2DM) I would suggest to use - Type 2 diabetes (T2D) everywhere in the text including the title. T2D is more common term.
  • instead using - diabetic patients and - diabetic subjects I would suggest to use patients with diabetes or subjects with diabetes
  • instead using - anti-diabetic agents I would suggest to use antihyperglycemic therapy or antihyperglycemic treatment
  • instead using - sugar control it should be written - glycemic control
  • instead using - HbA1C it should be written –HbA1c
  • instead using term hyperlipidemia I would suggest to use - dyslipidemia
  • in Table 1 there is a mistake in row - Hyperlipidemia, % 813? (81.3?)
  • furthermore, in Table 1 instead using fabric names Glucophage and Actos, I would suggest to use generic names of medication – metformin and pioglitazone

Author Response

Q1: Study limitations are listed clearly in the manuscript. However, I would suggest specifying study limitations in a separate paragraph.

Answer 1: We specify study limitations in as separate paragraph as your suggestion. Please see Page 8, “Discussion”, last paragraph.

Q2: English language and style are fine, however minor spell check is required. In the manuscript, I would suggest to change following terms:

instead using - Type 2 diabetes mellitus (T2DM) I would suggest to use - Type 2 diabetes (T2D) everywhere in the text including the title. T2D is more common term.

instead using - diabetic patients and - diabetic subjects I would suggest to use patients with diabetes or subjects with diabetes

instead using - anti-diabetic agents I would suggest to use antihyperglycemic therapy or antihyperglycemic treatment

instead using - sugar control it should be written - glycemic control

instead using - HbA1C it should be written –HbA1c

instead using term hyperlipidemia I would suggest to use – dyslipidemia

Answer 2: We correct these terms according to your suggestion. Please see words with blue color.

Q3: in Table 1 there is a mistake in row - Hyperlipidemia, % 813? (81.3?)

Answer 3: We correct it.

Q4: furthermore, in Table 1 instead using fabric names Glucophage and Actos, I would suggest to use generic names of medication – metformin and pioglitazone

Answer 4: We revise generic names of medication according to your suggestion.

Reviewer 3 Report

As a new finding, the authors relate a high qPCR expression of genes from microbiome species characteristic for the Firmicutes group to the severity of NAFLD in 163 persons with type 2 DM.

Scope: 

The paper is within the scope of JPM although not directly related to PM, the analyses of the microbiome are rather crude and analyses on a group basis.

Tile: No comments.

Abstract: Intro: NAFLD remains an important problem , not a has been. Results: Especially or only male

Key words: No comments.

Intro:  References to chronic infectious hepatitis (self-citations) out of scope. Fair introduction to the microbiome in tp 2

Methods: No comments. Fair to divide into groups of NAFLD, regression analysis OK.

Results: Logistic analysis is logistic regression analysis? Table 1: Avoid brand names (Actos, Glucophage).Table 2: Heading makes no sense. Figures OK. 

Disussion: Is fair, an attempt discuss the apparent paradox that  Firmicutes gene expression is expected to be low in Tp 2 DM is lacking, see e.g. PLOSone 2010 5: e9085. A critical evaluation of the reliability of the method (Fibroscan)would be welcome, I encourage the authors to study Nature Rev Gastroenterol and Hepatol 2020 7: 279.

Author Response

Point 1: Scope: The paper is within the scope of JPM although not directly related to PM, the analyses of the microbiome are rather crude and analyses on a group basis.

Response 1: Thanks for your comment. 

Point 2: Tile: No comments

Response 2: Thanks.

Point 3:Abstract: Intro: NAFLD remains an important problem , not a has been. Results: Especially or only male

Response 3: Thanks for your suggestion. We revise abstract as your suggestions

Point 4: Key words: No comments.

Response 4: Thanks

Point 5: Intro:  References to chronic infectious hepatitis (self-citations) out of scope. Fair introduction to the microbiome in tp 2

Response 5: We delete the references to chronic infectious hepatits (self-citations) as your suggestion

Point 6: Methods: No comments. Fair to divide into groups of NAFLD, regression analysis OK.

Answer 6: Thanks

Point 7: Results: Logistic analysis is logistic regression analysis? Table 1: Avoid brand names (Actos, Glucophage).Table 2: Heading makes no sense. Figures OK.

Response 7: This is logistic regression analysis. We revise the names of medication (metformin and pioglitazone) in Table 1 according to your suggestion. We revise the heading of Table 2 (The microbiome distribution of study participates)

Point 8: Disussion: Is fair, an attempt discuss the apparent paradox that Firmicutes gene expression is expected to be low in Tp 2 DM is lacking, see e.g. PLOSone 2010 5: e9085. A critical evaluation of the reliability of the method (Fibroscan) would be welcome, I encourage the authors to study Nature Rev Gastroenterol and Hepatol 2020 7: 279.

Response 8: Thanks for your suggestions. We revise the discussion and reference according to your comments. Please see Page 8, Discussion, first paragraph (reference 34) and fourth paragraph (reference 42,43).